# Exploring the Differential Impact of Salt Stress on Root Colonization Adaptation Mechanisms in Plant Growth-Promoting Rhizobacteria

**DOI:** 10.3390/plants12234059

**Published:** 2023-12-03

**Authors:** Lorena del Rosario Cappellari, Pablo Cesar Bogino, Fiorela Nievas, Walter Giordano, Erika Banchio

**Affiliations:** INBIAS Instituto de Biotecnología Ambiental y Salud, CONICET—Universidad Nacional de Río Cuarto, Campus Universitario, Río Cuarto 5800, Argentina; lcappellari@exa.unrc.edu.ar (L.d.R.C.); pbogino@exa.unrc.edu.ar (P.C.B.); wgiordano@exa.unrc.edu.ar (W.G.)

**Keywords:** PGPR, salt stress, biofilm, motility, autoaggregation capacity, *Mentha piperita*

## Abstract

Salinity inhibits plant growth by affecting physiological processes, but soil microorganisms like plant growth-promoting rhizobacteria (PGPR) can alleviate abiotic stress and enhance crop productivity. However, it should be noted that rhizobacteria employ different approaches to deal with salt stress conditions and successfully colonize roots. The objective of this study was to investigate the effect of salt stress on bacterial survival mechanisms such as mobility, biofilm formation, and the autoaggregation capacity of three plant growth-promoting strains: *Pseudomonas putida* SJ04, *Pseudomonas simiae* WCS417r, and *Bacillus amyloliquefaciens* GB03. These strains were grown in diluted LB medium supplemented with 0, 100, 200, or 300 mM NaCl. Swimming and swarming mobility were evaluated in media supplemented with 0.3 and 0.5% agar, respectively. Biofilm formation capacity was quantified using the crystal violet method, and the autoaggregation capacity was measured spectrophotometrically. In addition, we evaluated in vitro the capacity of the strains to ameliorate the effects of saline stress in *Mentha piperita*. The study found that the GB03 strain exhibited enhanced swarming mobility when the salt concentration in the medium increased, resulting in a two-fold increase in the halo diameter at 300 mM. However, high concentrations of NaCl did not affect the swimming mobility. In contrast, swimming motility was reduced in WCS417r and SJ04 under salt stress. On the other hand, exposure to 300 mM NaCl resulted in a 180% increase in biofilm formation and a 30% rise in the percentage of autoaggregation in WCS417r. Conversely, the autoaggregation percentage of the strains SJ04 and GB03 remained unaffected by saline stress. However, for GB03, biofilm formation decreased by 80% at 300 mM. Simultaneously, inoculation with the three evaluated strains alleviated the detrimental effects of salinity on plant growth. Under 150 mM salt stress, all strains showed increased fresh weight, with GB03 and WCS417r improving by 40% and SJ04 exhibiting the most remarkable effect with a 70% rise compared to non-inoculated plants. Despite their different strategies for mitigating salt stress, the application of these strains presents a promising strategy for effectively mitigating the negative consequences of salt stress on plant cultivation.

## 1. Introduction

Excessive salt in soil inhibits plant growth and development, leading to decreased crop yield [1]. The accumulation of sodium (Na^+^) in the root zone of plants can cause several deleterious effects, including alterations to soil physical–chemical properties. Soil salinity reduces soil organic matter content, soil water-holding capacity, and water infiltration and weakens soil structure and disrupts soil aggregate stability. Other common negative impacts of salinity on soil properties include increased soil pH, exchangeable sodium percentage (ESP), and sodium adsorption ratio (SAR), as well as reduced cation exchange capacity (CEC) and soil microbial community [2]. These effects restrict the absorption of water and usually cause ionic imbalances in plants [3,4]. Salt stress can also alter the levels of growth regulators, decrease protein synthesis, and impair photosynthetic processes in plants [5,6]. Additionally, salt stress can cause oxidative damage due to the accumulation of reactive oxygen species (ROS), which can be detrimental to cells at high concentrations [7,8,9]. If the intracellular concentration of ROS is not controlled, it can lead to damage to the cell structure by lipid peroxidation, protein oxidation, nucleic acid damage, and enzymatic inhibition, ultimately leading to cell death [10,11,12].

To mitigate salt stress, beneficial microorganisms known as plant growth-promoting rhizobacteria (PGPR) can play an important role. Rhizospheric bacteria can effectively colonize plant roots and maintain soil fertility by offering a favorable reinforcement to inorganic fertilizers and pesticides [13]. The promotion of growth in plants produced by PGPR can be diverse, resulting, for example, from the solubilization of phosphates, production of siderophores, biological fixation of nitrogen, production of the enzyme 1-aminocyclopropane 1-carboxylic acid (ACC) deaminase, production and regulation of phytohormones, biocontrol activity, production of volatile organic compounds (VOCs), and activation of induced systemic resistance (ISR) [14]. In the last decade, bacteria belonging to different genera (*Achromobacter, Azospirillum, Bacillus, Burkholderia, Enterobacter, Methylobacterium, Microbacterium, Paenibacillus, Pantoea, Pseudomonas, Rhizobium*, and *Variovorax*) have been reported to promote tolerance to various types of abiotic stress in plants, such as water, salt, and heat stress, as well as nutrient deficiency and heavy metals [15,16]. Several studies have shown that PGPR inoculation can improve plant growth and development under salt stress conditions [17,18,19]. The mechanisms behind these phenomena include osmotic adjustment, modulation of the plant antioxidant system, ion homeostasis, modulation of the phytohormonal balance, and increase in nutrient uptake [20]. For instance, our previous research showed that the microbial volatile organic compounds (mVOCs) produced by *Bacillus amyloliquefaciens* GB03 improve antioxidant status and ameliorate the effects of salt stress in *Mentha piperita* [21,22]. Similar effects were observed with mVOCs emitted by native *Pseudomonas putida* SJ46 and SJ04 isolated from the *Mentha piperita* rhizosphere [23,24]. Additionally, direct inoculation with *Pseudomonas simiae* WCS417r or *B. amyloliquefaciens* GB03 improved antioxidant status, leading to an enhancement of *M. piperita* plant growth under drought stress conditions [25]. However, the information available regarding the strategies employed by rhizobacteria to mitigate the negative effects of salt stress is limited [26].

Every living organism, whether it is unicellular or multicellular, possesses the inherent ability to respond to its environment [27]. Through the process of evolution, unicellular organisms have developed various mechanisms to effectively adapt and respond to changing environmental conditions [28]. Microbial motility is one of the most important features of microbial ecology and physiology [29,30]. The motility behaviors of bacteria enable them to search for nutrients but also represent an adaptive strategy to survive in harsh environmental conditions [27], such as saline soils. Flagella are necessary for both swimming and swarming motility [31]. Swimming is the typical motility in liquid media, while swarming occurs on semi-solid surfaces [31]. Bacteria within the genus *Pseudomonas* are able to move individually or collectively by swimming or swarming as a result of one or more polar flagella [32], while *Bacillus subtilis* possesses peritrichous flagella, which facilitate its motility [33]. Swarming is distinct from swimming since it requires secretions of a surfactant, an increase in flagellar density, and perhaps additional factors. The surfactant acts to reduce surface tension and create a thin layer of water within which to swarm [34,35]. The reason that cells require an increase in flagellar density is unknown, but it may be necessary to increase the total amount of thrust generated by the cell [34,35]. Moreover, biofilm formation and EPS production by PGPR are processes that help microorganisms endure adverse environments [36]. In saline environments, in particular, the high osmotic pressure usually causes cytoplasmic lysis and cell death. Hence, microorganisms develop biofilms, which protect them from such abiotic stress [37]. PGPR with EPS and biofilm production are better at colonizing plant root surfaces under unfavorable conditions [38]. In fact, biofilm formation is the most common strategy for bacterial life and survival in terrestrial habitats [39].

Biofilm formation and bacterial autoaggregation are two closely related processes that play crucial roles in bacterial adhesion and colonization on various surfaces, including biotic and abiotic surfaces [40]. Bacterial aggregations are highly specific processes that involve an interaction between surface molecules of microorganisms, which act as adhesins, and complementary receptors, including proteins and carbohydrates [41]. On the other hand, biofilm formation is a complex process that involves the attachment of bacteria to a surface, followed by the formation of a three-dimensional structure [42]. Autoaggregation plays a crucial role in the initial stages of biofilm formation by promoting bacterial adhesion and facilitating the formation of microcolonies [43]. Biofilms are multicellular communities embedded in a self-secreted extracellular matrix mainly composed of exopolysaccharides (EPSs) [44].

Epiphytic PGPR typically form biofilms once established in the root epidermis. In the case of *Bacillus subtilis*, biofilm formation is essential for root colonization, with the plant playing a crucial role in this interaction [45,46]. Particularly in *B. subtilis*, biofilm formation is facilitated by the extracellular matrix, which consists of two main components: an EPS (extracellular polymeric substance) and the protein TasA [45]. However, in the case of some *Pseudomonas* species, biofilm formation is not essential. Instead, they colonize the root as microcolonies, occupying the junctions between epidermal cells and are surrounded by plant-derived mucilage [47]. Therefore, different bacterial strains have different modes of arrangement in the roots, which can even vary depending on the type of plant being colonized [48].

As previously mentioned, microorganisms play a vital role in amending soil properties affected by saline soil and contribute to improving crop yield [49]. However, it should be noted that rhizobacteria employ different approaches to deal with salt stress conditions and successfully colonize roots. In this regard, the aim of this work was to evaluate different strategies of rhizobacteria to mitigate salt stress. Specifically, we analyzed the impact of salinity on the autoaggregation capacity, biofilm formation, and motility of three PGPR strains: *Pseudomonas simiae* WCS417r, *Pseudomonas putida* SJ04, and *Bacillus amyloliquefaciens* GB03. Additionally, we evaluated the capacity of plant growth promoted by these strains in salt-stressed *Mentha piperita* plants. These strains were evaluated based on their demonstrated capacity to enhance growth in various aromatic plant species under abiotic stress conditions by direct inoculation, as well as by the emission of VOCs [21,25,50,51,52]. However, it is not clear which strategies these strains use to adapt to adverse environments and promote plant growth.

## 2. Results

### 2.1. Effect of Salinity on Motility: Swimming and Swarming

After a 48 h incubation period, we assessed the motility of the different PGPR strains under salt conditions. All evaluated PGPR strains exhibited swimming motility, but only the WCS417r and SJ04 strains displayed a significant decrease when exposed to salt stress (*p* ≤ 0.05) (Table 1; Figure 1). This decrease became more pronounced with higher salt concentrations. In contrast, GB03 did not show any variation in swimming motility across the different salt concentrations evaluated (*p* > 0.05). However, for strain GB03, it was observed that as the concentration of salt in the medium increased, swarming-type mobility also increased. On the other hand, WCS417r and SJ04 presented low swarming mobility under the different salt concentrations evaluated (Table 1; Figure 1).

### 2.2. Autoaggregation Assay

Bacterial autoaggregation was quantified in different salt concentrations. All strains evaluated showed high levels of autoaggregation under control conditions, with 60% for WCS417r and SJ04 and 80% for GB03 (Figure 2). Under salt stress conditions, the autoaggregation values did not differ from the control for the SJ04 and GB03 strains. In contrast, WCS417r showed a 30% increase (*p* < 0.05) under the most stressful conditions (200 and 300 mM), reaching values of around 90%.

### 2.3. Effect of Salinity on Biofilm Formation

The ability to form biofilms has been observed to be influenced by exposure to salinity conditions in microorganisms. Therefore, the impact of salt on the formation of biofilms by different PGPR strains was explored. In Figure 3A, it can be observed that the growth of the WCS417r strain decreased considerably when it was exposed to high concentrations of NaCl (200 or 300 mM) (*p* ≤ 0.05). Although the growth of SJ04 decreased in response to salt stress, the decrease was not as abrupt as for the WCS417r strain. GB03 growth was not affected by exposure to low concentrations of NaCl (*p* > 0.05), and only GB03 growth was greater when the strain was exposed to 300 mM NaCl (*p* ≤ 0.05).

Figure 3B shows the values of absorbance at 570 nm as a measure of optical density (OD), which reflects the capacity of biofilm formation of the different PGPR strains evaluated in diluted (1:10) LB medium supplemented with different concentrations of NaCl (0, 100, 200, and 300 mM). The OD values for *P. simiae* WCS417r were significantly higher under salt stress conditions in the medium without added NaCl (0 mM) (*p* ≤ 0.05). This increase was almost 180% in relation to control conditions, and the values were similar regardless of the salt concentrations applied (Figure 3B). For SJ04, although the biofilm formation capacity increased when the strain was exposed to NaCl, the difference with respect to the control was not statistically significant (*p* > 0.05). In contrast, for GB03, this ability decreased significantly under saline stress as the treatment became more severe, being 40%, 76%, and 81% under 100, 200, and 300 mM, respectively (Figure 3B).

In spite of these observations about the capacity of biofilm formation, the biofilm/growth (B/G) (OD570/OD620) ratio can lead to more accurate conclusions about biofilm development since it is a normalized parameter for biofilm formation when the method of determination is based on CV measurements. The B/G ratio can be interpreted as a parameter of bacterial lifestyle under certain conditions. In the absence of stress, this ratio was higher for GB03 than for SJ04 and WCS417r (Figure 3C). Under saline stress, for GB03, the decrease in biofilm formation produced low B/G ratios; the higher the salt concentration, the lower the B/G ratio (*p* ≤ 0.05). In contrast, the B/G ratio for both *Pseudomonas* strains, the WCS417r and SJ04 strains, increased with higher values corresponding to higher salt concentrations (*p* < 0.05). The highest OD 570/OD620 ratio was observed for the WCS417r strain at 300 mM (ca. 34).

### 2.4. Effect of PGPR Inoculation on Plant Growth under Salinity Conditions

To determine the effects of different strain inoculations on peppermint plants subjected to saline stress, saline concentrations of 0, 100, and 150 mM were used. Preliminary trials were also conducted with higher concentrations (200 and 300 mM), and it was observed that the explants failed to develop at these concentrations.

Peppermint plants subjected to salt stress exhibited a reduction in their shoot fresh weight, as expected, resulting in a 25% decrease compared to the control plants (*p* < 0.05).

The inoculation with WCS417r, SJ04, and GB03, under non-salt stress conditions, significantly enhanced the fresh weight by approximately 36%, 29%, and 50%, respectively, compared to non-inoculated plants (Figure 4). When subjected to 100 mM salt stress, plants treated with WCS417r or SJ04 displayed a significant rise of 67–79% in comparison to non-inoculated stressed plants (*p* ≤ 0.05), while GB03 did not show any change in relation to the controls (*p* > 0.05). Under 150 mM salt stress, all tested strains exhibited an increase in fresh weight, with GB03 and WCS417r showing a 40% improvement, while the most remarkable effect was observed with SJ04, resulting in a 70% rise compared to non-inoculated plants (*p* ≤ 0.05).

## 3. Discussion

Soil salinity is a significant environmental stress that affects plant growth [5]. High concentrations of salts cause lowered water potential in the soil, which results in a wide range of physiological and biochemical alterations in plants. Salinity affects almost all aspects of plant development, including germination, vegetative growth, and reproductive development [15,53]. It imposes ion toxicity, osmotic stress, nutrient deficiency, and oxidative stress on plants, which collectively lead to impaired physiological processes, abnormal biochemical mechanisms, nutrient imbalance, abridged photosynthesis, altered enzymatic activities, loss of cell turgor, and accumulation of ROS [12]. These effects ultimately lead to reduced crop productivity and huge economic losses [15,54,55]. Not only does salt affect plant productivity, but it also affects rhizobacteria, modifying their phenotypic characteristics related to root colonization [56]. However, nature has provided microorganisms with mechanisms to sense their environments and respond accordingly. In bacteria, these responses can be characterized as sensory adaptations [57].

Swimming and swarming represent individual and collective cell motility [58,59]. It has been suggested that motility plays a significant role in the search for nutrients, interaction with the environment, and survival in adverse environmental conditions [60]. Moreover, motility appears to be crucial for biofilm formation [61]. This is because motility, which depends on flagella, fimbriae, and pili, allows cells to move and adhere to appropriate host surfaces. In turn, this facilitates colonization [62]. In the present study, all tested strains showed some degree of swimming and swarming ability under control conditions. However, when salt concentration increased, the swimming-type mobility decreased in *Pseudomonas* strains, and in *B. amyloliquefaciens* GB03, the swarming-type mobility increased. It has been reported that all strains that could form biofilm were also positive in terms of motility [63,64], suggesting that motility is critical for biofilm formation [65]. However, our results revealed that under salt stress conditions, strains that have weak biofilm ability showed high swimming motility and also some degree of swarming, while other strains with high biofilm formation ability showed poor motility. Papadopoulou et al. [66] observed similar results in *P. putida* strains. The strain that revealed weak biofilm ability showed high swimming motility and some degree of swarming, while the strain that exhibited a high degree of swarming and swimming motility was not able to form biofilm in vitro. These results, in agreement with our research, support the idea that the intensity of motility does not correlate with the capacity for biofilm formation and that other factors may be involved. For instance, the chemical composition, viscosity, and pH of the culture medium, as well as the incubation temperature, have been identified as potential contributing factors [67]. Additionally, attachment and subsequent biofilm formation can be influenced by various surface structures. Specifically, surface proteins have been identified to play a crucial role in the initial adherence to surfaces [68]. Nevertheless, despite the fact that individual motility was reduced, this was not an impediment to forming biofilm, which reinforces the idea that biofilm formation constitutes an important protection mechanism.

Autoaggregation is a mechanism by which bacteria can form aggregates or clumps, which can affect the capacity of root colonization [68].. Autoaggregation can promote bacterial adhesion to surfaces, including plant roots, and facilitate biofilm formation, playing a crucial role in the initial stages of biofilm formation [43]. According to the present study, the percentage of autoaggregation was only increased in *P. simiae* WCS471r at high salt concentrations. This suggests that the effect of salinity on autoaggregation may depend on the bacterial species and environmental conditions [69]. A positive correlation was found between autoaggregation and biofilm formation in some bacterial species, including *Sinorhizobium meliloti*, but the correlation may not always hold true for other bacterial species [42]. In the present study, we observed a correlation between autoaggregation and biofilm formation only in WCS417r when exposed to 200 and 300 mM NaCl. Unfavorable growth conditions or low metabolic activity have been found to induce aggregative behavior in bacteria that normally grow in a dispersed, non-aggregated manner. In this context, autoaggregation may reflect a survival strategy that is triggered under hostile environmental conditions [68].

Biofilm-forming PGPR have been found to exhibit a number of beneficial features, such as facilitating resource acquisition and higher resistance to antibiotics and adverse environmental stresses (e.g., high temperature, extreme pH, salinity, and drought), and as a result, their chances of survival in competitive soil environments are enhanced [69,70,71,72]. Biofilm production is a significant characteristic that enables PGPR to withstand various environmental stresses and maintain a high cellular presence attached to plant roots. This allows them to exert their beneficial interactions in the rhizosphere [73]. Furthermore, biofilm formation not only protects PGPR from stressful conditions but also directly benefits plants [74]. Biofilm formation is a multistep process that requires the integration of various bacterial physiological processes, including quorum sensing [75,76], motility [64,77], autoaggregation [42], and EPS production [78,79]. The results of the present study revealed that biofilm formation was variable for the evaluated strains. Both *Pseudomonas* strains showed higher biofilm formation under salinity stress conditions, with similar results being registered for *P. stutzeri* MJL19, a bacterial strain isolated from the rhizosphere of halophyte plants [80]. It is important to note that in WCS417r and SJ04, biofilm production was promoted under high concentrations of NaCl (200 and 300 mM), despite the fact that this concentration inhibited bacterial growth, especially in the case of WCS417r (Figure 3A). This result supports the idea that biofilm can be associated with a protection mechanism, allowing bacteria to survive and thrive in environments containing high salt concentrations. Biofilm development by rhizobacteria is expected to enhance rhizosphere competence both under normal and stress conditions [69,81]. Biofilm-forming rhizobacteria are expected to colonize plant roots more stably and can remain active more consistently. Moreover, exopolysaccharides have water-retaining and adhesive characteristics and play a significant role in the stabilization of soil aggregates and the regulation of nutrients and water flow across plant roots [82]. *B. amyloliquefaciens* GB03 exhibited a decrease in biofilm formation as the salt concentration in the medium increased. However, its growth remained unaffected by exposure to NaCl. Related to this, other studies have demonstrated that salt-tolerant bacteria can survive in various salt concentrations and counteract the effects of salt through different mechanisms, including the accumulation of osmoprotectants, compatible solutes, and specialized extracellular polymeric substances (EPSs), which can contribute to improved plant growth in saline soil [83,84]. In agreement with this, Ansari et al. [56] found that *Bacillus subtilis* FAB10 displayed improved biofilm development when exposed to lower concentrations of NaCl treatment. However, at higher concentrations, there was a notable decrease in biofilm formation compared to the control.

Previous research has reported that increased NaCl concentration can reduce bacterial EPSs and biofilm formation [56]. However, plant growth-promoting rhizobacteria with EPS-producing characters can chelate different cations, including Na^+^, and bind with Na^+^ ions through the secretion of EPSs, which reduce their toxicity in the soil. Therefore, a large population of EPS-producing bacteria in the rhizosphere zone can reduce the concentration of available Na^+^ for plant uptake and alleviate salt stress effects on plants in a saline environment. This suggests that halotolerant biofilm-producing rhizobacteria can enhance crop productivity under saline agro-ecosystems by decreasing the accumulation of harmful Na^+^. EPSs under salinity stress can bind sodium ions and alleviate their toxic effect in the soil [85,86]. Biofilm formation in moderately halophilic bacteria is also believed to be helpful for plant growth improvement in saline soil [13,87], as we observed in inoculated *M. piperita* plants grown under 100 or 150 mM NaCl. When the plants were exposed to 100 or 150 mM, the fresh weight of peppermint plants that were not inoculated decreased, while, when plants were inoculated indistinctly with WCS417r, SJ04, or GB03, a significant increase in the fresh weight of the aerial parts was observed compared to their corresponding controls. Remarkably, both the GB03 and the *Pseudomonas* strains exhibited an ability to mitigate the negative effect of salinity, despite having distinct mechanisms of adaptation to saline environments. Similar results were observed in *M. piperita* plants inoculated with *P. simiae* WCS417r under drought stress [25]. In previous studies, we were able to determine that *M. piperita* plants grown in a saline medium and exposed to GB03 VOCs had significantly better morphological characteristics and higher total chlorophyll contents compared to control plants. In addition, they showed a reduction in malondialdehyde (MDA) levels, which are considered to be an indicator of lipid peroxidation and membrane damage, and had an increased antioxidant capacity in relation to plants cultivated under salt stress but not treated with mVOCs. Through chromatographic analysis, it was determined that the percentage of acetoin emission increased when the bacterium was exposed to NaCl [21]. These results demonstrate the potential of GB03 mVOCs to diminish the adverse effects of salt stress [22]. In a related study, Vaishnav et al. [88] observed an up-regulation of salt-tolerant gene expression in soybean plants that were inoculated with the biofilm-producing PGPR *P. simiae* strain AU under salinity stress conditions. Thus, biofilm-producing bacteria may possess advantages over other types and be able to thrive in adverse environments and deliver beneficial effects to plants.

## 4. Materials and Methods

### 4.1. Bacterial Strains and Culture Conditions

The three strains, previously reported as PGPR, *Pseudomonas simiae* WCS417r (formerly known as *P. fluorescens* WCS417r [89]), *Bacillus amyloliquefaciens* GB03 (originally described as *B. subtilis* GB03 [90]), and *Pseudomonas putida* SJ04 (GenBank KF312464.1), were isolated from the rhizosphere of *M. piperita* from a commercial plantation in the Villa Dolores Region, Córdoba, and used for the bacterial cultures [23]. Bacteria were grown on Luria–Bertani (LB) medium [91] (10 g/L tryptone, 5 g/L yeast extract, 5 g/L NaCl) for routine use and maintained in nutrient broth with 15% glycerol at −80 °C for long-term storage. For the different assays, the strains were grown in diluted (1:10) LB medium and supplemented with 0, 100, 200, or 300 mM NaCl.

### 4.2. Motility: Swarming and Swimming

The swimming and swarming of the WCS417r, GB03, and SJ04 strains were evaluated in the presence of different concentrations of NaCl (0, 100, 200, and 300 mM). Swimming motility was determined in plates containing diluted (1:10) LB medium supplemented with corresponding concentrations of NaCl (0, 100, 200, or 300 mM) and 0.3% *w*/*v* agar. A bacterial suspension was inoculated by puncture in the center of the plate. Subsequently, the plates were inverted and incubated for 48 h at 28 ± 2 °C, and the colony diameters were measured in centimeters (cm) [92].

For the swarming assay, different concentrations of NaCl were added to diluted (1:10) LB medium containing 0.5% *w*/*v* of agar. A quantity of 2 μL of an inoculum grown in LB medium for 24 h at 28 ± 2 °C was placed at the center of the agar surface and incubated for 48 h at 28 ± 2 °C. The swarming extent was determined by measuring the swarming diameters (cm) and comparing them with the corresponding controls [93]. This assay was repeated three times, and each experiment contained ten replicates.

### 4.3. Autoaggregation Assay

The autoaggregation capacity of the strains was determined as described by Sorroche et al. [42]. The bacteria were grown in LB medium overnight at 30 °C with rotation at 120 rpm until the stationary phase and were later centrifuged at 10,000 rpm for 10 min at 25 ± 2 °C and then resuspended in diluted (1:10) LB medium supplemented with the corresponding concentration of NaCl to attain an optical density OD620 nm of 1. Bacterial suspensions (5 mL) were then transferred to a glass tube (10 × 70 mm) and allowed to settle for 24 h at 4 °C. A 200 µL aliquot of the upper portion of the suspension was transferred to a microtiter plate, and the OD600 was measured (OD final) with a MicroELISA Auto Reader (Series 700 Microplate Reader; Cambridge Technology). A control tube was vortexed for 30 s, and the OD600 was determined (OD initial). The autoaggregation percentage was calculated as 100× (OD final/OD initial). This assay was repeated three times, and each experiment contained ten biological replicates.

### 4.4. Biofilm Formation Assay

Biofilm formation was determined macroscopically by a quantitative assay using 96-well microtiter dishes made of polyvinyl chloride. Biofilms were stained with crystal violet (CV) based on the method of O’Toole and Kolter [94], with modifications. The bacteria were grown in LB medium overnight at 30 °C with rotation at 120 rpm until the stationary phase and were later centrifuged at 10,000 rpm for 10 min at 25 °C and then re-suspended in an appropriate volume of diluted (1:10) LB medium supplemented with the corresponding concentration of NaCl (0, 100, 200, or 300 mM) to achieve an initial OD 620nm of 0.1. A quantity of 150 µL of each bacterial suspension was added to each well and incubated for 24 h at 28 ± 2 °C. Bacterial growth was quantified by measuring the absorbance of planktonic cells in each well at OD620 with a MicroELISA Auto Reader (Series 700 Microplate Reader; Cambridge Technology). Planktonic cells were gently aspirated with a pipette, each well was washed 3 times with saline solution (0.9% *w*/*v* NaCl), and cells adhered to the polystyrene support were stained with 180 μL of a CV aqueous solution (0.1% *w*/*v*) for 15 min. The wells were rinsed repeatedly with distilled water, and biofilm formation was assayed by the addition of 150 μL of 95% *v*/*v* ethanol. The OD570 of solubilized CV was measured with a MicroELISA Auto Reader, as described above. In parallel, sterile control cultures were established in diluted (1:10) LB medium supplemented with the corresponding concentration of NaCl. Relative BFA (biofilm-forming ability) was calculated as OD570/OD620 (biofilm quantified by staining with CV relative to planktonic growth measurement) [68]. This assay was repeated three times, and each experiment contained ten biological replicates for each treatment and control.

### 4.5. Plant Micropropagation

Young shoots from *Mentha piperita* plants grown in the Traslasierra Valley (Cordoba Province, Argentina) were surface disinfected by soaking for 1 min in 17% (*v*/*v*) sodium hypochlorite solution and then rinsed three times in sterile distilled water. The disinfected shoots were cultured in 100 mL Murashige and Skoog culture medium (MS) [95] containing macro- and micronutrients, vitamins, 0.66 mg/L indolebutyric acid (IBA), 0.025 mg/L naphthalene acetic acid (NAA), 0.7% (*w*/*v*) agar, and 3% (*w*/*v*) sucrose). The pH was adjusted to 5.6–5.8. After 30 days, apical meristems with foliar primordia that did not show contamination were aseptically removed from the terminal buds, and new explants were cultured in test tubes in 40 mL MS medium [23]. Plantlets obtained from tips were multiplied by single-node culture. Explants were grown in a growth chamber under controlled conditions of light (light/dark cycle: 16/8 h), temperature (22 ± 2 °C), and relative humidity (ca. 70%).

### 4.6. Inoculation Assay

The bacterial culture was grown in LB medium overnight at 30 °C with rotation at 120 rpm until the stationary phase and later centrifuged at 10,000 rpm, washed twice in 0.9% *w*/*v* NaCl by centrifugation (10,000 rpm, 10 min, 25 ± 2 °C), resuspended in sterile water, and adjusted to a final concentration of ca.10^9^ colony-forming units cfu/mL for use as an inoculum.

Glass bottles (250 mL) coated with MS (0.5% % *w*/*v* agar) were inoculated with 100 μL bacterial suspension (10^8^ cfu/bottle) and with sterile water for the treatment control. After solidification of the medium, one node from an aseptically cultured plant was placed in the center of the bottle. Distilled water was used as a negative control.

The glass bottles containing plants and bacteria were covered with aluminum foil, sealed with parafilm to avoid contamination, and placed in a growth chamber under controlled conditions of light (16 h/8 h light/dark cycle), temperature (22 ± 2 °C), and relative humidity (ca. 70%).

Salt stress was generated by the addition of NaCl. Different concentrations of salt were added to the MS media (plant growth media) during its preparation, including 0 mM, 100 mM, and 150 mM NaCl. A solution of 1000 mM NaCl was used to achieve the desired final NaCl concentration. The medium was then sterilized by autoclaving [96]. Previously, a test was carried out to determine the maximum concentration of salt that the plants could withstand. This assay was repeated three times, and each experiment contained ten biological replicates (10 bottles per treatment; 1 plant/bottle). After 30 days, the plants were harvested, each plant was removed from its glass bottle, and the shoot fresh weight was measured.

### 4.7. Statistical Analyses

The data were subjected to an analysis of variance (one-way ANOVA) followed by a comparison of multiple treatment levels with controls using Fisher’s post hoc LSD (least significant difference) test. Differences between means were considered significant for *p* values ≤ 0.05. The Infostat software program, version 2020 (Group Infostat, Universidad Nacional de Córdoba, Argentina), was used for all the statistical analyses.

## 5. Conclusions

The present study revealed that the three evaluated PGPR strains have different strategies for mitigating salt stress. Mobility plays an important role in the survival of microorganisms under adverse conditions. The three strains studied exhibited swimming and swarming motility under control conditions. Under saline stress, only *B. amyloliquefaciens* GB03 increased swarming motility as a strategy to colonize new areas in search of more favorable conditions. *Pseudomonas* may utilize alternative adaptive mechanisms. On the other hand, autoaggregation capacity and biofilm formation represent strategies associated with the generation of protective structures in stress situations. In this context, when subjected to salt stress conditions, both the *P. simiae* WCS417r and *P. putida* SJ04 strains exhibited an enhancement in biofilm formation, while only the *P. simiae* WCS417r strain demonstrated an increase in the percentage of autoaggregation at high NaCl concentrations. Notably, a correlation between autoaggregation and biofilm formation was observed specifically for the *P. simiae* WCS417r strain at high concentrations of NaCl in the medium. However, all the evaluated PGPR strains were able to adapt to salt stress through the different mechanisms of biofilm formation, autoaggregation, and mobility, suggesting that these strains could be useful in promoting the survival, and therefore the colonization, of plants grown under salt stress conditions, which was reflected in the observed plant growth promotion of *M. piperita.* Thus, inoculation with these strains could represent a strategy for mitigating the effects of salt stress in plants grown under these conditions. However, with the aim of increasing agricultural productivity, future research should now focus on exploring the mechanisms of cross-talk between PGPR and plants in salinity stress environments in order to obtain a better understanding of the mechanisms by which PGPR and plants interact in adverse conditions, and also on investigating the importance of several adaptive mechanisms in plants toward salt stress.

## Figures and Tables

**Figure 1 plants-12-04059-f001:**
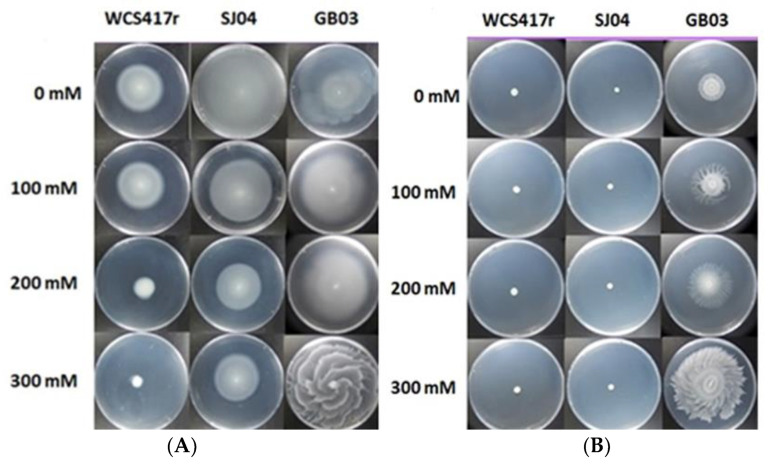
Effect of exposure to different concentrations of NaCl on the swimming (**A**) and swarming (**B**) motility of the strains *P. simiae* WCS417r, *P. putida* SJ04, and *B. amyloliquefaciens* GB03.

**Figure 2 plants-12-04059-f002:**
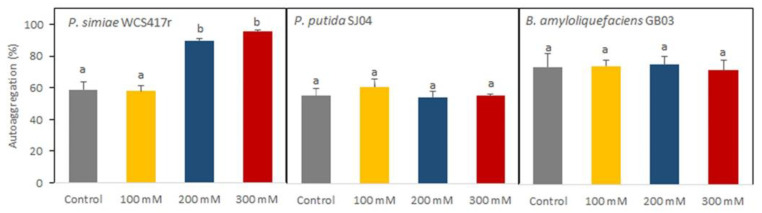
Percentage of autoaggregation of the strains *P. simiae* WCS417r, *P. putida* SJ04, and *B. amyloliquefaciens* GB03 under salt stress conditions (0 mM, 100 mM, 200 mM, and 300 mM). Values are means ± standard errors (SEs) of three independent replicates (*n* = 3). Means followed by the same letter in a given column are not significantly different according to Fisher’s LSD test (*p* ≤ 0.05).

**Figure 3 plants-12-04059-f003:**
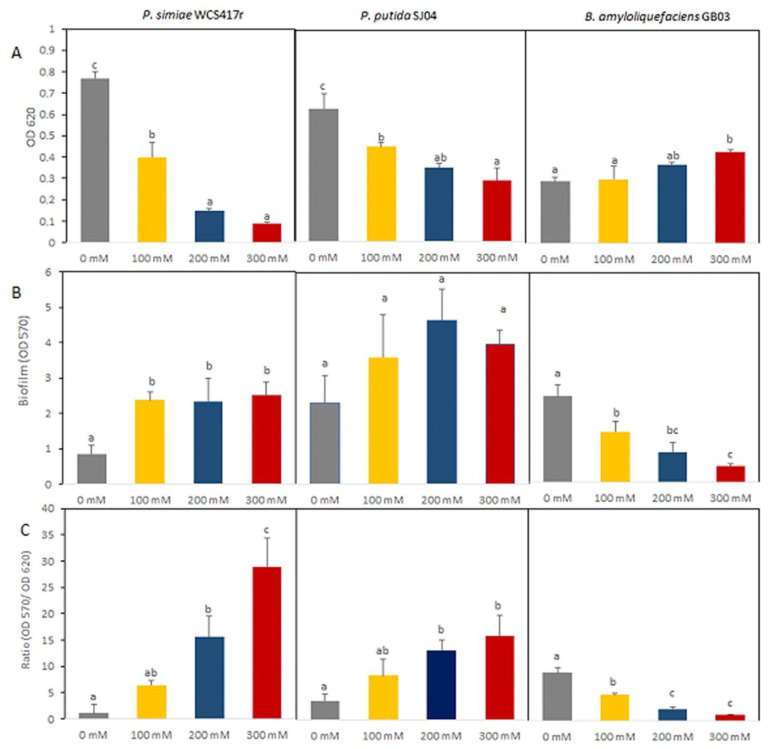
Biofilm formation of the strains *P. simiae* WCS417r, *P. putida* SJ04, and *B. amyloliquefaciens* GB03 under saline conditions (0, 100, 200, and 300 mM). (**A**) Growth of strains under different salt concentrations (OD). (**B**) Biofilm development on polystyrene surfaces shown as an average of OD measurements at 570 nm. (**C**) The relation between biofilm and growth is shown as the ratio between ODs at 570 and 620 nm. Values are means ± standard errors (SEs) of three independent replicates (*n* = 3). Means followed by the same letter in a given column are not significantly different according to Fisher’s LSD test (*p* ≤ 0.05).

**Figure 4 plants-12-04059-f004:**
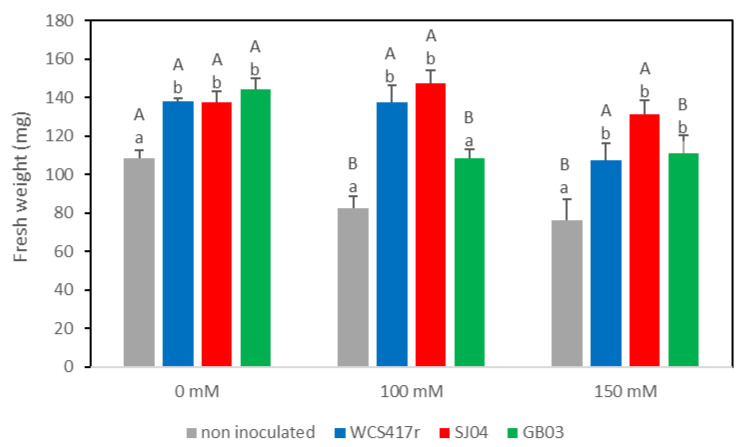
Effects of inoculation with different PGPR strains (*P. simiae* WCS417r, *P. putida* SJ04, and *B. amyloiquefaciens* GB03) on fresh weight of *Mentha piperita* plants grown in Murashige–Skoog medium with 0, 100, and 1500 mM NaCl. Values are means ± standard errors (SEs) of three independent replicates (*n* = 3). Values with different letters (capital letters for the same strains at different NaCl concentrations and lowercase letters for different strains at the same NaCl concentration) denote significant differences among treatment groups according to Fisher’s LSD test (*p* ≤ 0.05).

**Table 1 plants-12-04059-t001:** Diameters of swimming and swarming of the strains *P. simiae* WCS417r, *P. putida* SJ04, and *B. amyloliquefaciens* GB03 grown under different salt concentrations (means ± SEs). Means followed by the same letter within a column are not significantly different according to Fisher’s LSD test (*p* ≤ 0.05).

Strain	NaCl	Swimming (cm)	Swarming (cm)
WCS417r	0mM	5.05 ± 0.12 a	0.70 ± 0.06 a
	100 mM	5.50 ± 0.25 a	0.71 ± 0.08 a
	200 mM	3.86 ± 0.32 b	0.74 ± 0.06 a
	300 mM	1.56 ± 0.23 c	0.70 ± 0.07 a
SJ04	0 mM	7.84 ± 0.21 a	0.48 ± 0.06 a
	100 mM	6.58 ± 0.23 b	0.52 ± 0.08 a
	200 mM	4.83 ± 0.32 c	0.52 ± 0.04 a
	300 mM	3.12 ± 0.17 d	0.48 ± 0.07 a
GB03	0 mM	6.99 ± 0.32 a	2.78 ± 0.14 a
	100 mM	7.38 ± 0.19 a	3.49 ± 0.17 b
	200 mM	6.34 ± 0.34 a	3.95 ± 0.18 b
	300 mM	6.48 ± 0.68 a	6.07 ± 0.24 c

## Data Availability

The data presented in this study are available on request from the corresponding author. The data are not publicly available due to privacy restrictions.

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
