# Peer review of "Exploring the Differential Impact of Salt Stress on Root Colonization Adaptation Mechanisms in Plant Growth-Promoting Rhizobacteria"

_plants, 2023, doi:10.3390/plants12234059_

Round 1

Reviewer 1 Report

Comments and Suggestions for Authors

The present article  entitled “Exploring the Differential Impact of Salt Stress on Root Colonization Adaptation Mechanisms in Plant Growth-Promoting Rhizobacteria” based on interesting  theme and author have tried to explored the potential of PGPR in salt stress management. During the experiment authors have performed very limited experiment , so need  some  explanation.

.Authors have selected the strains on the basis of PGP traits, biofilm formation and autoaggregation capacity, and just after inoculation they have analysed their plant growth.

Do the author have  checked , the impact of inoculation on the native microflora of the plant,

Do the author have used any marker that  confirm  inoculant strains enter to the plants and regulates plant growth.

Comments on the Quality of English Language

The present article  entitled “Exploring the Differential Impact of Salt Stress on Root Colonization Adaptation Mechanisms in Plant Growth-Promoting Rhizobacteria” based on interesting  theme and author have tried to explored the potential of PGPR in salt stress management. During the experiment authors have performed very limited experiment , so need  some  explanation.

.Authors have selected the strains on the basis of PGP traits, biofilm formation and autoaggregation capacity, and just after inoculation they have analysed their plant growth.

Do the author have  checked , the impact of inoculation on the native microflora of the plant,

Do the author have used any marker that  confirm  inoculant strains enter to the plants and regulates plant growth.

Author Response

Reviewer #1: Comments and Suggestions for Authors

The present article entitled “Exploring the Differential Impact of Salt Stress on Root Colonization Adaptation Mechanisms in Plant Growth-Promoting Rhizobacteria” based on interesting theme and author have tried to explored the potential of PGPR in salt stress management. During the experiment authors have performed very limited experiment , so need some explanation.
.Authors have selected the strains on the basis of PGP traits, biofilm formation and autoaggregation capacity, and just after inoculation they have analysed their plant growth.

Do the author have checked , the impact of inoculation on the native microflora of the plant,

-We appreciate the reviewer's inquiry regarding the impact of inoculation on the native microflora of the plant. It is necessary to note that the primary objective of our work was to study the mechanisms of adaptation to salt stress of PGPR strains. While the interaction between introduced PGPR strains and the native microflora is indeed an important aspect of microbial ecology, it was beyond the scope of this particular study.
However, we acknowledge the significance of understanding how inoculated strains may affect the indigenous microbial communities associated with the plant. Such interactions can influence plant growth, health, and stress resilience. Future studies could certainly be designed to investigate the effects of inoculation on the native microflora, which would contribute valuable insights into the impact of PGPR application in agricultural systems.

Do the author have used any marker that confirm inoculant strains enter to the plants and regulates plant growth.

-In response to the reviewer's query, while specific markers confirming the entry of inoculant strains into plants and their regulatory effects on plant growth were not employed in the current study, our prior research provides compelling evidence of the positive impact of these inoculants on plant health and growth [21,22, 23, 24].
In previous studies, we have examined the direct effect of both direct inoculation and the volatile organic compounds (VOCs) emitted by these strains on mint plants, under normal conditions and under abiotic stress (saline and water stress). Overall, it was observed that inoculation with these strains mitigated the negative effects of abiotic stress, as evidenced by reduced malondialdehyde (MDA) levels, an indicator of lipid peroxidation and membrane damage. Additionally, there was an increase in antioxidant capacity, as indicated by DPPH (2,2-diphenyl−1-picrylhydrazyl) radical scavenging activity, and other evaluated parameters.
However, it is important to note that the specific mechanisms by which these strains enter plants remain subjects of interest for future investigations. We acknowledge the need for further exploration into these mechanisms in subsequent studies.

21. Cappellari, L.R.; Banchio, E. Microbial Volatile Organic Compounds Produced by Bacillus amyloliquefaciens GB03 Ameliorate the Effects of Salt Stress in Mentha piperita Principally Through Acetoin Emission. J. Plant Growth Regul. 2019, 39, 764–775. https://doi.org/10.1007/s00344-019-10020-3
22. Cappellari, L.R.; Chiappero, J.; Palermo, T.B.; Giordano, W.; Banchio, E. Volatile Organic Compounds from Rhizobacteria Increase the Biosynthesis of Secondary Metabolites and Improve the Antioxidant Status in Mentha piperita L. Grown Under Salt Stress. Agronomy 2020, 10, 1094. https://doi.org/10.3390/agronomy10081094
23. Santoro, M.V.; Bogino, P.C.; Nocelli, N.; Cappellari, L.R.; Giordano, W.F.; Banchio, E. Analysis of Plant Growth-Promoting Effects of Fluorescent Pseudomonas Strains Isolated from Mentha piperita Rhizosphere and Effects of Their Volatile Organic Compounds on Essential Oil Composition. Front. Microbiol. 2016, 7, 1085. https://doi.org/10.3389/fmicb.2016.01085
24. Gil, S.S.; Cappellari, L.R.; Giordano, W.; Banchio, E. Antifungal Activity and Alleviation of Salt Stress by Volatile Organic Compounds of Native Pseudomonas Obtained from Mentha piperita. Plants 2023, 12, 1488. https://doi.org/10.3390/plants12071488

Reviewer 2 Report

Comments and Suggestions for Authors

In section 2.4, "Effect of PGPR Inoculation on Plant Growth under Salinity Conditions," I believe it would be beneficial to note that the control treatment exhibits statistically significant differences at various NaCl concentrations. This could serve as a crucial point to indicate that the NaCl concentration is indeed causing noteworthy alterations in the control treatment. Consequently, any observed changes in the other treatments can be attributed to the bacteria, thereby confirming the proper execution of the plant trial.

In section 4.2, Motility: Swarming and Swimming, it is indicated that the method for inoculating the plates for swarming and swimming tests is different. Is there any reason for this?

Line 375/388 does not specify the growing conditions required for bacteria cultivation until reaching the stationary phase (time, temperature, agitation(rpm)).

Line 420, growing conditions (time, temperature, agitation(rpm)).

Line 432, how and when is the salt added to the media?.

General: I believe that performing an additional analysis of colony-forming units for each bacterium at different NaCl concentrations would further substantiate the notion that these bacteria can withstand NaCl-induced stress through different strategies, as indicated in the conclusion (Line 447). For future experiments, I also suggest that investigations into salt or water stress should not only involve measuring fresh weight but also include dry weight measurements. This approach would enable conclusions about the actual increase in organic matter in the plant and the plant's utilization of water.

Author Response

Reviewer #2:

Comments and Suggestions for Authors

In section 2.4, "Effect of PGPR Inoculation on Plant Growth under Salinity Conditions," I believe it would be beneficial to note that the control treatment exhibits statistically significant differences at various NaCl concentrations. This could serve as a crucial point to indicate that the NaCl concentration is indeed causing noteworthy alterations in the control treatment. Consequently, any observed changes in the other treatments can be attributed to the bacteria, thereby confirming the proper execution of the plant trial.

-We apologize for the oversight in our initial submission. We appreciate your suggestion and agree that it is crucial to highlight the statistically significant differences at NaCl concentrations in the control treatment in section 2.4, "Effect of PGPR Inoculation on Plant Growth under Salinity Conditions."

We have now included a more detailed statistical analysis in our study. This analysis discriminates the effects of inoculation with the same strain at different NaCl concentrations, as well as the effects of different strains at the same NaCl concentration. This allows us to clearly demonstrate the negative impact of salinity on plant growth, which was inadvertently omitted in our initial submission.

In this analysis, values with different letters (capital letters for the same strains at different NaCl concentrations and lowercases for different strains at the same NaCl concentration) denote significant differences among treatment groups according to Fisher’s LSD test (p ≤ 0.05).

This additional analysis further supports our findings and confirms the role of the bacteria in plant growth and stress tolerance under different NaCl concentrations.

In section 4.2, Motility: Swarming and Swimming, it is indicated that the method for inoculating the plates for swarming and swimming tests is different. Is there any reason for this?

-The methods for inoculating the plates for swarming and swimming tests are indeed different, and this is due to the distinct nature of these two types of bacterial motility.

Swimming motility is typically assessed in a low-viscosity environment, such as a semi-solid agar medium with a low agar concentration (around 0.3% w/v), which allows flagellated bacteria to move through the medium. The inoculation by puncture in the center of the plate allows the bacteria to move outwards from a single point, and the resulting colony diameter can be measured to assess the swimming ability of the bacteria.

On the other hand, swarming motility occurs on a more viscous surface, such as a higher agar concentration (around 0.5% w/v), which is still moist but offers more resistance to movement. Swarming is a more complex behavior that often involves groups of cells moving together and can be influenced by surface wetness and nutrient availability. The inoculation for swarming assays is done by placing a small volume of bacterial culture on the surface of the agar, allowing the bacteria to spread across the surface as a coordinated group.

These differences in the inoculation method reflect the different conditions required to facilitate and observe each type of motility. Swimming assays are designed to allow individual cells to move freely in a semi-solid medium while swarming assays are designed to observe the collective movement of bacterial populations on a solid surface.

Line 375/388 does not specify the growing conditions required for bacteria cultivation until reaching the stationary phase (time, temperature, agitation(rpm)).

-The information has been added to the manuscript

Line 420, growing conditions (time, temperature, agitation(rpm)).

-The information has been added to the manuscript

Line 432, how and when is the salt added to the media?.

-We apologize for any lack of clarity regarding the addition of salt to the media. The salt was incorporated into the Murashige and Skoog (MS) medium at the time of its preparation. Specifically, we used a 1000 mM NaCl solution to achieve the desired final NaCl concentration in the medium. Following the addition of salt, the medium was then sterilized by autoclaving. We have now included this detail in the manuscript.

General: I believe that performing an additional analysis of colony-forming units for each bacterium at different NaCl concentrations would further substantiate the notion that these bacteria can withstand NaCl-induced stress through different strategies, as indicated in the conclusion (Line 447). For future experiments, I also suggest that investigations into salt or water stress should not only involve measuring fresh weight but also include dry weight measurements. This approach would enable conclusions about the actual increase in organic matter in the plant and the plant's utilization of water.

Specifically in the study of biofilm formation, when growth is determined through spectrophotometry at OD 620 (Fig 3a), the effects of salt on strain survival can be inferred. While we are aware of the limitations of the technique, it enables us to approximate survival under these conditions.

The utilization of optical density (OD) measurement in our study serves as a practical and widely accepted method for assessing bacterial growth. OD measurement through spectrophotometry provides a means to quantify the turbidity of the culture medium, which is indicative of cell density. This method is particularly valuable for tracking changes in bacterial populations over time and under different experimental conditions

We will take this recommendation into account for our future research and ensure that both fresh and dry weight measurements are incorporated into our experimental design to enhance the robustness of our findings. Thank you for the constructive feedback, which will undoubtedly improve the quality of our research outcomes.

Round 2

Reviewer 1 Report

Comments and Suggestions for Authors

Author have made significant changes during revision. The article can be accepted in the current form.

Comments on the Quality of English Language

Author have made significant changes during revision. The article can be accepted in the current form.